# Membrane Separation Used as Treatment of Alkaline Wastewater from a Maritime Scrubber Unit

**DOI:** 10.3390/membranes12100968

**Published:** 2022-10-02

**Authors:** Maryse Drouin, Giulia Parravicini, Samy Nasser, Philippe Moulin

**Affiliations:** 1Aix Marseille Univ, Centrale Marseille, CNRS, M2P2, EPM, 13331 Marseille, France; 2CMA Ships, Boulevard Jacques SAADE, 4 Quai d’Arenc, CEDEX 02, 13235 Marseille, France

**Keywords:** marine closed-loop scrubber wastewater, ultrafiltration, silicon carbine membrane, exhaust gas cleaning systems, backflushing action

## Abstract

Since 1 January 2020, the sulfur content allowed in exhaust gas plume generated by marine vessels decreased to 0.5% m/m. To be compliant, a hybrid scrubber was installed on-board, working in closed loop and generating a high volume of alkaline wastewater. The alkaline water suspension was treated by a silicon carbide multitubular membrane to remove pollutants, and to allow the water discharge into the natural environment. In this paper, membrane filtration behavior was analyzed for the maritime scrubber wastewater. A range of operating parameters were obtained for several feedwater quality-respecting industrial constraints. The objective was an improvement of (I) the water recovery rate, (II) the filtration duration, and (III) the permeate quality. Thus, in high-fouling water, a low permeate flow (60 L h^−1^ m^−2^) with frequent backflushing (every 20 min) was used to maintain membrane performance over time. In terms of water quality, the suspended solids and heavy metals were retained at more than 99% and 90%, respectively. Other seawater discharge criteria in terms of suspended solids concentration, pH, and polyaromatic hydrocarbons were validated. The recommended operating conditions from laboratory study at semi-industrial scale were then implemented on a vessel in real navigation conditions with results in agreement with expectations.

## 1. Introduction

Maritime transport represents one of the most efficient modes of large-scale transportation and plays a fundamental role in the world market trade, especially for its economic interdependence. It has been estimated that maritime transportation accounts for more than 80% of the world market. For instance, the Suez Canal Authority revealed that 5303 vessels used its shipping lane from January to the end of March 2022, representing an incremental increase of 15.8% compared to 2021 [1]. International shipping accounts for more than 15% of nitrogen oxides (NO_x_) emissions, approximately 10% of sulfur oxides (SO_x_) and almost 8% of particulate matter in total global emissions [2,3,4]. However, it is responsible for a proportion of less than 3% of the total anthropogenic CO_2_ emissions [5]. Thus, to limit marine, air, and water pollution, the International Maritime Organization (IMO) adopted stricter emission regulations for maritime vessels. In the past decade, international rules have been adopted to reduce the sulfur emission from ships’ plumes. Regulations are listed in Annex IV of the International Convention for the Prevention of Marine Pollution from Ships known as the MARPOL Convention [6]. On 1 January 2020, the sulfur concentration allowed in exhaust gas plumes was reduced from 3.5% to 0.5% worldwide and even down to 0.1% in sulfur emission controlled areas (SECA) [7,8]. This important energetic transition, known as Cap Sulfur 2020, represents a major challenge for shipowners. Many studies have been made to satisfy the limits defined by the MARPOL Convention regarding SO_x_ reduction [9,10,11]. The compliant option proposed is the exploitation of exhaust gas cleaning systems (EGCS) also known as scrubber units directly installed in line with exhaust gas piping on maritime vessels [12,13].

Exhaust gas cleaning systems represent the most attractive option; the operational costs are reduced and counterbalance the investment costs for retrofitted vessels [12,14,15] According to Clarkson’s World Fleet Register (November 2019), nearly 3000 scrubbers have already been installed on existing ships, which corresponds to around 3% of the total number of operational vessels and supplementary vessels that are currently in retrofit for this technology. Moreover, usage of scrubber units largely reduces ship pollution emissions. It has been reported that seawater scrubber units commonly remove 90–95% of SO_2_ content in exhaust gas, 10–20% of NO_x_, along with 80% of particulate matter and around 10–20% of hydrocarbons [16,17]. When exhaust gas cleaning systems are operated in closed loop (CL), mainly in port and coastal areas, the ship impact on marine biodiversity, environment, and population is significantly reduced [18,19]. Indeed, port and coastal areas are defined as sulfur emission control areas, where stricter regulations are set in terms of water discharge and SO_x_ atmosphere emission (0.1% S instead of 0.5% S) [7,8,20]. In CL configuration, a certain volume of scrubbing water is continuously recirculated to the absorption column, causing a pollutant accumulation and the acidification of wastewater. Indeed, when engines’ exhaust gases meet the alkaline water, the gaseous molecules of sulfur oxides (SO_x_) are converted to sulfurous acid ions (SO_2_^3−^) and sulfuric acid ions (SO_4_^2−^), which causes the acidification of the scrubbing suspension and a decrease in gas treatment efficiency [12,21]. An alkaline compound is added to process water to improve the gas pollutant removal by neutralizing its acidic pH. Magnesium hydroxide (MgOH_2_) is preferred as the alkaline agent for flue-gas desulfurization on board for its nontoxic and high metal adsorption properties [22,23,24]. Additionally, mixed with seawater, 90% of gaseous SO_2_ can be removed [25]. The CL wastewater is sent to a wastewater treatment unit to decrease the particulate matter, hydrocarbons, and heavy metal concentration of process tank water and allows the water to be discharged into the sea. The residual water produced, highly concentrated with suspended solids and pollutants, is stored in a residue tank on board and unloaded when the ship arrives in the port. The volume of residue tanks is sometimes very low, which limits the CL operation capability for a few days. In this study and for the first time, marine exhaust gas desulfurization units have been coupled with a membrane process as a water treatment unit. Membrane filtration is well known as being efficient for complex effluent water treatment. It has already been used in many applications for wastewater treatment [26,27,28] because it allows the limiting the toxic compounds and pollutants from wastewater and produces a good permeate quality regardless of the water quality variation.

Some articles present in the current literature report a membranes process used to treat marine effluent directly on board due to their large specific surface areas [29,30,31]. Accordingly, membrane filtration processes have been reported as the most efficient process for oily wastewater treatment. For example, Tomczak and Gryta [32] reported that it is possible to obtain a permeate without any trace of oil and with a reduction of 80% of organic compounds, thanks to ceramic membrane treatment. The discharged water satisfied the environmental regulations which require a maximum oil and grease concentration of 15 mg L^−1^. Additionally, membrane filtration processes coupled with pretreatment steps have been reported to be efficient to remove suspended solids, turbidity, and heavy metals from wastewater. For instance, Abdullah et al. [33] presented a good retention rate with heavy metals such as cadmium, mercury, lead, and chromium by using membrane filtration after coagulation and complex metal formation pretreatment steps. Any research reported the membrane performances for marine scrubbers’ water treatment. Moreover, some studies deal with the usage of membrane technology, mainly with a succession of treatment steps from ultrafiltration (UF) to reverse osmosis (RO) for flue-gas desulfurization onshore [34,35,36,37,38]. In case of ultrafiltration applied to this field, researchers report good, suspended solid removal and, according to Yin et al. [35], UF membrane permeability stabilization is around 130 L h^−1^ m^−2^ bar^−1^.

The choice of membrane material is determined by their properties regarding the treated suspension. In the field of membrane filtration, ceramic membranes are well established to treat wastewater containing a high proportion of organic matters mainly due to their properties [39,40]. Furthermore, compared to conventional mineral ceramic membrane materials, silicon carbide (SiC) membranes present the highest permeability (>3000 L h^−1^ m^−2^) [41] due to their very low tortuosity, their good chemical resistance, and their mechanical strength [42]. SiC membranes are currently used in multiple applications, such as drinking water, heavy metal removal, food, and biotechnology treatment [43], and microalgae production [44]. Regarding oil and grease treatment, SiC membrane removal efficiency was demonstrated by Das et al. [45] for produced water treatment in which they had an oil rejection between 89% and 94% from an initial feed water of 1.557 mg L^−1^ oil concentration. A suspension turbidity reduction of 94% was obtained. These studies confirmed the choice of installing membrane filtration separation to treat the scrubber water mainly composed of natural salty water, hydrocarbons, heavy metal, particulate matter, and unburned fuel residue.

The novelty of this paper shows the membrane filtration process being studied for the first time as an alternative process to treat scrubber wastewater in the maritime field from a semi-industrial plant to an industrial-boarded scale plant. Membrane processes can reach significant levels, well below the water discharge criteria, and can be easily adapted to the various feedwater quality. Moreover, membrane separation units have a compact design, and low operational costs [46]. In terms of membrane performances, the high SiC membrane permeability allows one to maintain filtration for longer periods of time, which can satisfy the navigation and effluent storage constraints. Indeed, as is shown by Hofs et al. [40], the same membrane fouling can be obtained with higher permeate flux applied on a SiC membrane surface in comparison to other membrane materials. This is why it is important to study membrane operation performances for treating maritime scrubber water and its industrial applications.

In this context, this study’s aims are: (I) to observe the behavior of SiC membranes installed on marine vessels under different operative conditions as permeate flux, filtration cycle duration, backflush action, and water quality, (II) to define the best operating parameters that satisfy industrial marine constraints such as low concentrate volume produced, the longest filtration time, good permeate water quality (hydrocarbons, turbidity, pH, and heavy metal have been taken into consideration), and (III) to compare semi-industrial scale results with the onboard membrane filtration for parameter validation and study the membrane process flexibility during current ship navigation.

To perform the study, five effluents sampled from container vessels and representative of the entire scrubber water fleet variability were filtered by SiC membranes on a semi-industrial scale. Several operating parameters are applied to the membrane for each fluid characteristic with the objective of defining the best parameter for an onboarded application. First, regarding the filtration tendencies and physical and chemical analysis made on water, effluents were categorized from high- to low-fouling capacity. Then, the impact of each parameter was studied in each fluid category (from high- to low-fouling). A range of operating parameters were obtained for high- and low-fouling fluid. To finish, parameters defined for high-fouling fluid were applied on the onboarded unit to validate the results and compare the filtration behavior.

## 2. Materials and Methods

### 2.1. Pilot Plant and Membrane Description

To evaluate sustainable operating conditions for filtration onboard, a semi-automatic membrane filtration pilot plant (Figure 1) was designed to emulate the real membrane process installed onboard. The filtration was carried out in cross-flow circulation mode. The filtered water was continuously sent to the permeate tank, and the concentrate water was only eliminated during backflushing. Respectively, onboard, the permeate water was sent to a storage tank before being discharged into the environment. The concentrate water was stored in a residue tank before being discharged once the ship arrived at the nearest port awaiting further specific treatment onshore. For example, and to highlight the significance of this study, onboard, the feed flow of the membrane unit was around 8 m^3^ h^−1^ for an available residue tank volume between 85 and 150 m^3^, even less in some ships, which limits the number of closed loop scrubbers’ days in operation between two residue tanks draining: 8–15 days respectively for a high recovery rate of 95%.

Throughout the membrane filtration experiments, the inlet pressure was fixed at 1.5 bars and the fluid circulation was maintained at a turbulent regime on membrane channels (Re = 7500), with a constant velocity at 2.5 m s^−1^. To emulate the operating conditions present onboard, filtration tests were carried out at a constant permeate flow rate, which represents an input variable of the system. Due to the high-fouling tendency of membranes, a semi-autonomous backflush (BF) operation was performed to limit the irreversible membrane fouling and to maintain the SiC membrane filtration performance over time. Two modes of backflush actions were defined: backwash (BW) and backpulse (BP). The backwash action was divided into two phases: first the injection of water was accompanied by a permeate pressure rise, and then the injection of water under fixed pressure of 3 bars throughout the desired time interval. Backpulse action worked in the same way as the BW but with shorter duration. The permeate water was injected through the membrane only when a permeate pressure of 3 bars was reached.

SiC membranes employed in this study were supplied by LiqTech (Liqtech International, Hobro, Denmark), and have been reported to be applied advantageously in processing industrial wastewater [44]. The membrane used has a multichannel configuration, with 30 cylindrical channels of 3-mm diameter each, with a total length equal to 1178 mm and a total active area equal to 0.33 m^2^. The average pore size was defined to be equal to 0.2 µm. The clean water permeability was reported to be equal to 3200 L h^−1^ m^−2^ bar^−1^. This value was taken into consideration when evaluating the membrane permeability recovery after the chemical cleaning of each filtration test.

### 2.2. Experimental Tests and Analyses

Filtration tests were made in batch mode at constant permeate flow with a continuous recirculation of the water suspension in the filtration membrane loop. SiC membrane performance was estimated by evaluating the increase of the irreversible resistance (R_irr_) generated on the membrane after each backflush operation and in opposition to the reversible resistance (R_rev_) (i.e., the variation of permeability over time) removed by a physical cleaning action. The total resistance was defined as the sum of the irreversible, reversible, and intrinsic resistance of the membrane. R_irr_ and R_rev_ can be fully removed by chemical cleaning. To investigate the filtration efficiency over time, two recovery rates were defined: the total filtration water recovery rate (R_w_) calculated, including the total volume lost during BF operation, and the filtration permeate recovery rate (R_filtr_), maintained at a high value, close to 100% for all experiments.

Samples of permeate and concentrate were taken before and after every BF to determine their physical and chemical characteristics of turbidity, conductivity, pH, dry matters (DM) and dissolved metals. The turbidity of each sample was measured by using a turbidimeter (WTW Lab Turbidity Meter Turb^®^ 550 IR, Xylem Analytics, Weilheim, Germany). Due to the high differences in turbidity present in treated effluents, concentrate samples and feed samples were diluted by a factor of 50 due to the suspension opacity and aggregation [35] and not to be above the detection limit. Complementary analyses such as conductivity were measured by a conductimeter profiline 3100 (tetra cond. 325 sensors, Xylem Analytics, Germany), and a pH analysis with a pHmeter HANNA HI 2221 with HI 1121 (HANNA Instrument, Woonsocket, RI, USA) sensor and dry matter analysis (from standard NF EN 12880) were made. Heavy metal concentration in permeate and concentrate samples were determined by spectrometry (ICP/MS) in a certified laboratory (Laboratoire Phytocontrol Waters, Nîmes, France) according to standards NF EN ISA 15587-2 and NF EN ISO 17294-2. A panel of eight metals were measured, these being lead (Pb), cadmium (Cd), arsenic (As), aluminum (Al), chrome (Cr), nickel (Ni), vanadium (V) and zinc (Zn). It was noticed that vanadium and nickel metal are the main metals found in the burned fuel of marine transportation vessels, and thus their concentrations have been reported to be significant on samples analyzed [47]. At the end of each test, a chemical cleaning procedure was performed (alkaline and acidic batch). Water permeability was recorded after each step to evaluate the membrane cleaning efficiency.

### 2.3. Effluents

Feedwaters were collected directly from the exhaust gas treatment closed loop recirculation tanks (CL process tank) present onboard to evaluate the variability of process water characteristics. A preliminary study, not presented in this paper, of scrubber water characteristics showed a significant difference in water quality depending on the type of vessels studied and, on the localization, where the sampling took place. Water suspensions treated were sampled during European navigation routes which took place between the Algeciras and Hamburg ports. Their main differences consisted of, in addition to their composition, the type of bunkered fuel burned, localization, and the type of engine considered in each ship. At least two different types of engines are installed in transportation ships—the main engine (ME) and the auxiliary engines (AE). Whereas the main engine is responsible for the ship’s propulsion, auxiliary engines are used for electrical power production onboard (which can represent up to 15% of the total fuel consumption). After an extensive internal study of the effluent’s characteristics (more than 50 samples collected from all over the world) and to overcome the large variability of real effluents treated by water treatment units installed onboard, five representative fluids were chosen from different ships and different engine process tanks (ME or AE). Their physical properties are given in Table 1. Because the scrubber waters onboard are usually pretreated by coagulation and the hydro-cyclone process before being sent to the filtration unit, the results obtained in a semi-industrial scale study underestimate SiC membrane performance due to a higher presence of suspended matter.

## 3. Results and Discussion

### 3.1. Membrane Performances Overview

The variability of feed water quality on membrane performance was studied by using five effluents coming from operational EGCS-coupled membrane filtration process retrofitted vessels (Table 1). A large range of operating conditions were applied based on the previous results obtained for each effluent and water quality. As examples, low- and medium-fouling fluid properties were filtered with a permeate flux higher than 150 L h^−1^ m^−2,^ whereas on the high-fouling water properties, permeate flux of 90 and 60 L h^−1^ m^−2^ were imposed on the membrane. Additionally, almost all BF operating conditions were tested on the three water categories (low-, medium-, and high-fouling). Regarding the BF duration, 20 s and 5 s were mainly used for BW action and 5 s for a BP injecting respectively 13, 6, and 2 L from the permeate side through the membrane. The BF frequency applied during filtration tests varied between 20 and 60 min. The lower BF frequencies (20–40 min) refer to BP actions in order to compensate the lower water volume injected and limit the irreversible fouling on membrane surface. As a consequence, a varied range of membrane performance in terms of fouling behavior was obtained for each water quality (Figure 2).

Effluents treated have been categorized by their fouling properties, depending on the water quality and permeability variation observed throughout the filtration time. High-fouling fluids were defined as the effluents from APL VANDA ME (V-ME) and CC KERGUELEN (KERG). For theses effluents, a higher dry matter (DM ≈ 87 and 127 g L^−1^) and suspended solid concentration (TSS ≈ 1.15 and 0.55 g L^−1^) were reported, and a lower permeability range was observed, lower than 400 L h^−1^ m^−2^ bar^−1^ (Figure 2d,e). APL SINGAPURA ME (S-ME) and APL VANDA AE (V-AE) waters were considered as a low-fouling fluid. Indeed, the operating permeability measured for S-ME was higher (around 600 L h^−1^ m^−2^ bar^−1^ (Figure 2a) and suspended solid concentration was lower (0.4 g L^−1^) in comparison to other effluent results. Regarding APL SINGAPURA AE effluent (S-AE), it was described as medium-fouling effluent (Figure 2c).

Due to (I) the high permeability of silicon carbide membranes, (II) the fluids’ physicochemical characteristics, and (III) the operating conditions applied on the permeate side, reverse flux effects can be observed on the membrane outlet. This phenomenon was mainly noticeable during the first filtration cycles, during which the pressure measured in permeate was higher than the membrane module outlet pressure (P_permeate_ > P_outlet_). Under this condition, the permeate volume produced is lower than the total volume of water filtered by the membrane, which limits the membrane fouling and sometimes overestimates performance. The reverse flow proportion decreases when the permeate pressure decreases enough with a constant outlet pressure. This results in a higher membrane active area fouling and a rapid decrease in permeability. Similar membrane filtration tendencies have already been noticed by Ghidossi et al. [48] and Springer et al. [49] with difficulties in measuring the initial permeability when the membrane was cleaned. The reverse flow in the SiC membrane throughout the filtration was also observed in other applications, as in food and beverages [43,50]. Regarding the results obtained, a high reverse flow was mainly observed for a low-fouling effluent. For instance, the permeate pressure measured throughout the filtration of S-ME water at 150 L h^−1^ m^−2^ permeate flow was in the same order as the pressure outlet. (Figure 3a). This pressure range explained the higher permeability mentioned around the time, 600 L h^−1^ m^−2^ bar^−1^. In case of high-fouling water filtration, a high concentration of fouling particles in water limited reverse filtration flow in the first minutes of each cycle, reducing the range of membrane permeability. For instance, throughout the filtration of KERG fluids shown in Figure 3b, permeate pressure was quite similar as the outlet pressure during the first filtration cycles due to a dilution effect. Thus, the permeability measured was higher than 400 L h^−1^ m^−2^ bar^−1^. After 40 min of filtration, a high membrane-fouling appears, the permeate pressure rapidly decreases with the permeability values, and no more reverse flow is observed on the membrane side.

### 3.2. Filtration Repeatability

Performed scale filtration with real feedwater implied that one verified the effluent quality in case of variability over time. In addition, two filtration tests were carried out, at the beginning and the end of the campaign (after two months) with a permeate flux of 90 L h^−1^ m^−2^. A BW was triggered every 40 min with a duration of 5 s under pressure on the membrane surface. The turbidity of the feed sample was measured at the beginning of each filtration test to evaluate the water quality. The values obtained were in the same range of turbidity, respectively equal to 200 and to 190 NTU for test 1 and test 2. Membrane permeability variations versus filtration time are shown in Figure 4a, where curves obtained are stackable. Indeed, permeability differences noticed after 170 min of filtration can be explained by an increase of turbidity present in the membrane loop after regulation deviation. Through the filtration test 2, a lower volume of clear water was injected during the BW actions. An injection of 5 L for test 2 vs. 5.5 L for test 1 implies a higher turbidity concentration factor in the loop (Figure 4b). In conclusion, filtration made on the same membrane was considered repeatable, allowing the results to be compared in this study.

### 3.3. Impact of Permeate Flow

The permeate flow was kept constant during filtration, as it is imposed on industrial applications. According to the fluid quality and fouling properties previously determined, the influence of permeate flux on filtration performances was studied for the S-ME and KERG water, respectively, a low- and a high-fouling property fluid. For KERG scrubber wastewater, three permeate flows were applied to the same membrane—150, 90, and 60 L h^−1^ m^−2^, with similar BW conditions (1 BW 5 s/40 min). The application of a permeate flow equal to 150 L h^−1^ m^−2^ (J150) on SiC membrane implies a strong permeability decline from the first minutes of filtration, as is shown in Figure 5. Thus, a permeate flow around 150 L h^−1^m^−2^ was not reported as viable for a long-term filtration. Regarding other operating conditions, the decrease of the permeate flow from 90 to 60 L h^−1^ m^−2^ (J90 and J60) with a similar feed water composition allows a lower permeability value stabilization, respectively equal to 250 and 90 L h^−1^ m^−2^ bar^−1^. This phenomenon is mainly linked to membrane properties such as high water permeability and to the pilot regulation system. In case of J60 experiments, the permeate flow rate required was too low for unit regulation. A transition phase with a lower permeate pressure was measured (larger gap between pressure outlet and pressure permeate) and appears to strongly reduce the permeability values (Figure 5c). In contrast, with a higher permeate flow, the permeate pressure was higher, and the filtration behavior was positively impacted by reverse flow on the membrane. The fouling resistance supports this idea: a lower permeate flow produces higher membrane-fouling mainly due to the pressure ratio (Figure 5b). Indeed, the reversible and irreversible fouling resistance created increased more rapidly throughout the filtration time when the permeate flow was lower. Additionally, the global water recovery rate was lower, around 70% and 77% for, respectively, J60 and J90, and a larger concentrate volume was generated in comparison to the permeate volume produced. Experiment results showed that filtration was maintained for a longer time.

In comparison with low-fouling water, such as S-ME water, applying a permeate flow higher than 150 L h^−1^ m^−2^ produced higher permeability value stabilization—approximately 250 L h^−1^ m^−2^ bar^−1^, as is shown in Figure 6. However, a lower water recovery rate of 67% was obtained. On the same effluent, implementing a permeate flow from 150 to 225 L h^−1^ m^−2^ increased the total filtration resistance and membrane fouling (Figure 6). For a permeate flow of 225 L h^−1^ m^−2^, a higher reversible resistance was observed, which highlighted an increased BW efficiency. The main part of the generated membrane-fouling was the reversible one. Thus, despite the higher feed turbidity of 177 NTU, an important part of suspended solids present on membrane surfaces were removed during physical cleaning actions. In this situation, a higher permeability recovery after each BW sequence was observed, followed by a rapid decrease throughout the filtration cycle until a lower value of approximately 270 L h^−1^ m^−2^ bar^−1^ was reached. In comparison, for a permeate flow of 150 L h^−1^ m^−2^ the final permeability cycle stabilization was approximately 400 L h^−1^ m^−2^ bar^−1^. From the results obtained, using a permeate flow of 225 L h^−1^ m^−2^ with physical cleaning every 40 min, allowed us to maintain the filtration for many hours and reduced the concentrate volume produced, with a water recovery rate reaching 80%.

Fluid properties influenced the membrane filtration performance. According to fluid characteristics and fouling properties considered, the optimum operating conditions are different mainly in terms of permeate flow applied. Indeed, in low-fouling water, a good filtration permeability was maintained with a high permeate flow, and a larger volume of water was treated. For example, on S-ME water, a permeate flow of 250 L h^−1^ m^−2^ was maintained with a high permeability value of 400 L h^−1^ m^−2^ bar^−1^. In comparison with high-fouling water, a permeate flow value of 150 L h^−1^ m^−2^ cannot be applied for more than 2 h without complete membrane fouling. From these results, treated scrubbers process water, with a permeate flow less or equal to 90 L h^−1^ m^−2^, seems a good compromise for onboarded filtration. Additionally, it is supposed that in cases of high-fouling water, using a permeate flow of 90 L h^−1^ m^−2^ with harder backflush conditions could be too stressful for the membrane; thus, decreasing the permeate flow to 60 L h^−1^ m^−2^ would be more appropriate. A lower permeate flux helps to increase the filtration time by reducing the membrane fouling. Under these conditions chemical cleaning frequency can be reduced. This is the first time that permeate flow values were prescribed for the treatment of exhaust gas cleaning system wastewater. The division into three types of effluent may appear simple, but it brings a simplified operation onboard. It has already been the case in other fields such as wine [43].

### 3.4. Impact on Backwashing Operating Duration

Backwashing is applied on the membrane surface to reduce the fouling. Water injection in countersense of the filtration helps to remove the fouling layer and reduce the chemical cleaning frequency. From an industrial point of view, it can be interesting to reduce the chemical consumption and increase the operation time. Water injection time during a backwashing action was reduced from 20 s to 5 s in order to evaluate the impact on the water recovery rate, fouling removal efficiency and membrane performance. According to BW definition, reducing the BW injection time significantly decreased the high constant pressure injection duration with a similar transition time (required to obtain the BW pressure). Filtration was made under similar conditions for different fluid quality. After considering previous results, permeate flows of 150 and 90 L h^−1^ m^−2^ were set, respectively, on S-ME and V-ME wastewater to prevent membrane fouling. In the case of high-fouling fluid (V-ME), filtration results (Figure 7) highlighted an increase in water recovery rate when the BW duration was reduced to 15 s. The water recovery rate also increased from 53% to a value higher than 75%, which is the minimum requirement for industrial applications. In terms of membrane-fouling behavior, except for the first filtration cycle (40 min), permeability and reversible fouling resistance were in the same order for both filtration (Figure 7). Moreover, a low permeability recovery was noticed after each physical cleaning action linked to the high particle deposit on the membrane side. However, injecting more water was not a solution to obtain a better BW efficiency, and the reversible resistance was low and similar for both conditions tested, below 4.10^11^ m^−1^ (Figure 7b).

In cases of low-fouling water treatment (S-ME water, Figure 8), similar observations were made. The water recovery rate increased from 67% for 1 BW 20 s/40 min to 80% for 1 BW 5 s/40 min; thus, the membrane filtration behavior was impacted. Additionally, it was found that the reversible fouling resistance formed was reduced when the BW volume decreased. The measurement of the reversible resistance for each filtration cycle was approximately 2.10^11^ and 3.5.10^11^ m^−1^ when the BW duration was 5 s and 20 s, respectively. Thus, a part of the backwashing volume injected for a 20-s duration was not helpful to remove the membrane-surface fouling. Consequently, under these conditions, a permeability stabilization around 500 L h^−1^ m^−2^ bar^−1^ was observed. Applying a shorter BW time also influenced the loop turbidity ratio, and membrane fouling. The turbidity ratio measured in filtration (turb_i_/turb_feed_, turb_i_ is the turbidity of the sample, turb_feed_ is the turbidity of the feed water) was around 10 for a 5-s BW, and only 6 for a 20-s BW. Thus, for an average feed, water turbidity of 145 NTU and the same filtration time, the suspended particle concentration was reduced when more water was injected. In both cases, the backwashing action was sufficient to remove the fouling layer on the membrane surface.

From the results presented, it was observed that injecting more than 6 L of water into the permeate, linked with a BW duration of 5 s, did not help in removing fouling layers on the membrane side for both fouling water qualities. Additionally, it was assumed that only half the water volume was needed to remove the fouling layer when a BW duration of 20 s was used, and 12 L were injected. In conclusion, increasing the BW duration did not impact the membrane filtration. This observation has already been demonstrated by Ye at al. [51] and confirmed by Slimane et al. [52] for seawater ultrafiltration. They increased the BW duration and BW frequency to their maximum value, and no membrane fouling reduction was observed. The value of 5 L was the minimum value of the optimal volume range (5–10 L) given by Slimane et al. [52], are in agreement since the SiC membranes used in this study have larger permeability. That is why it is preferable to decrease the BW duration in order to increase the water recovery. From an industrial point of view, an increase in the recovery rate helped to reduce the volume of residue water produce. Because residue tank storage is a critical point for onboarded filtration, we reduce the volume of water sent to this tank increase the day of CL running. That is why these results are important.

### 3.5. Impact of Backwashing Operation Frequency

The BW frequency refers to the delay between two successive reverse permeate water injections on the membrane side. Decreasing the backwash frequency led to a higher filtration cycle. The stress applied to the membrane surface was maintained over a longer period; consequently, a greater fouling was observed, and the irreversible fouling proportion increased more rapidly over time. Filtration cycles of 40 and 60 min, and BW durations of 5 s were applied to low- and high-fouling water (respectively, S-ME and KERG waters). The permeate flow was adjusted as a function of the feedwater quality, according to previous results. The filtration was realized with a higher permeate flow rate of 150 L h^−1^ m^−2^ flow for S-ME water, and a lower permeate flow of 90 L h^−1^ m^−2^ was applied to the membrane for KERG water experiments. In both cases, previous estimations were validated (Figure 9 and Figure 10). For KERG water filtration, lower permeability and higher irreversible fouling resistance values were observed when the BW interval was increased from 20 min (Figure 9). For 60-min filtration intervals, it was noticed that after the third filtration cycle, the permeability of the membrane dropped, and it became completely fouled. The membrane was not able to maintain the permeate flow at the desired value. Usage of BW allows a brief recovery of flow rate permeate, which validates its efficiency to limit the membrane-fouling over time during a short filtration duration. Filtrating at 90 L h^−1^ m^−2^ with a high BW interval limited the concentrate volume produced, and thus a water recovery rate of more than 80% was obtained. However, a higher membrane-fouling was reported, and the filtration run was stopped after 3 h; this condition was not sustainable in the long term.

In the cases of S-ME scrubber wastewater filtration, when the filtration cycle was increased from 40 to 60 min, the total fouling resistance increased faster (Figure 10). A higher irreversible fouling resistance was generated during filtration; thus, on the membrane side, a large portion of particles were not removed by BW. Indeed, the turbidity ratio before and after permeate injection was only decreased by 1.5 units. Nevertheless, filtration was maintained around the time and a permeability stabilization higher than 400 L h^−1^ m^−2^ bar^−1^ was observed. Similarly, KERG water filtration results, showed that increasing the BW interval helps to obtain a higher permeate recovery rate, 10% higher with 60-min intervals whereas 40-min intervals reached a percentage rate of 88%.

A similar filtration tendency was observed for both fluid characteristics. Increasing the BW-triggered delay increased the membrane fouling and irreversible resistance created on the membrane surface. These results are in agreement with Ye et al.’s [51] studies for seawater in hollow membrane filtration. They have shown that a more compact fouling cake was produced when the filtration cycle increased. However, increasing the filtration time allows obtaining a higher recovery rate. Weschenfelder et al. [53] concluded that usage of BF actions helped to increase the permeate flux, but its drawbacks were about the permeate water loss throughout time. Indeed, because the BF are made with permeate water, they triggered more BF in the same duration to generate a global water recovery rate reduction. This could be an important issue for onboarded operation due to the higher concentrate volume produced.

### 3.6. Impact of Backflushing Type: Backwash vs. Backpulse

Both physical cleaning actions, BW and BP, were performed on the membranes during the filtration of water coming from KERG. A backflush duration of 5 s was applied to the membranes every 40 min; thus, permeate water was injected with a fixed pressure during 5 s whereas the permeate flow was maintained at 90 L h^−1^ m^−2^. Usage of BP allowed for the reduction of the volume injected from 5 L (BW volume) to 2 L in comparison to BW which increased the permeate recovery rate from 77% to 90%. Filtration curves showed a lower permeability value when BP was used (Figure 11a), but a higher membrane fouling was generated. In fact, the filtration loop turbidity remained high due to the low volume of permeate water injected during backpulsing. The turbidity ratio (turb_i_/turb_feed_) measured after 150 min of filtration was 5 units when BP was used, whereas 4 units were calculated for BW actions. The higher suspended solids concentration on the membrane side, with BP, also produced the higher irreversible fouling observed (Figure 11b). The reversible fouling resistance shown in Figure 11b represents the BP efficiency in fouling removal, and filtration performance maintenance for several hours. In comparison, BW shows good performance, but it is less attractive for industrial applications due to the higher injected volume which gives a lower water recovery rate of 77%.

In the case of KERG water filtration, the influence of permeate flow and BW parameters have been discussed previously. Results have shown that applying 60 L h^−1^ m^−2^ as a permeate flow allows one to maintain the membrane filtration by limiting the filtration reverse flow perturbations thanks to the regulation parameter and an initial greater fouling. BF mode and frequency of applications were studied in order to define the sustainable and optimized physical cleaning operating conditions for a permeate flow of 60 L h^−1^ m^−2^. Filtration with 5 s BW every 40 min has already been discussed and was compared to other filtration tests performed with 5 s BP every 20 and 40 min, respectively (Figure 12). A high initial membrane fouling and a low permeability value stabilization less than 100 L h^−1^ m^−2^ bar^−1^ over time were observed for each filtration (Figure 12a). The physical cleaning action was efficient, the BF permeate water injection helped to reduce the fouling, maintain filtration, and a high reversible fouling resistance was achieved, as is shown by Figure 12b. For the same filtration interval, usage of BP instead of BW had no impact on fouling tendencies, as was seen in the permeability curves and irreversible fouling value (Figure 12). However, during backpulsing a lower volume of permeate was reinjected (1.7 L against 5 L for BW action) on the membrane side. Consequently, the reversible resistance generated was reduced, and the water recovery rate increased to a value of more than 80%. Applying BP more often in the membranes (1 BP every 20 min) reduced the irreversible fouling resistance in comparison to the other BW or BP conditions tested with 60 L h^−1^ m^−2^ as permeate flow and a high reversible fouling resistance were noticed (Figure 12b). For the same condition (5 s BP/20 min) a high permeability recovery in the beginning of each cycle was observed, accompanied by a significant decrease right after the restart of filtration. A loss of 100 L h^−1^ m^−2^ bar^−1^ in 20 min of filtration was noticed. Nevertheless, this condition seems ideal because a constant permeability drop during the filtration cycle and throughout the time was observed.

For low- and medium-fouling water, initial results on V-AE water filtration shows low water recovery rates around 50% and 65% (Figure 13b), respectively. Thus, permeate flow was increased up to 250 L h^−1^ m^−2^, and BP actions were applied to the membrane with the objective of concentrate volume limitations, and a strong increase in the water recovery rate. First, under these permeate flow conditions, the permeate water recovery rate increased to a value higher than 75%, increasing in the same way as the membrane fouling (Figure 13). Then, different BF conditions were applied to permeate flow of 250 L h^−1^ m^−2^. Results indicated that the usage of 5 s BP actions induced 2 L of permeate water used during backpulsing, which was six times less than the volume used during a 20-s BW. Thus, even if BP actions were executed more often on the membrane, as in every 20 min, the total volume of water loss was reduced, which significantly increased the water recovery rate to a value higher than 90%. In this condition (250 L h^−1^ m^−2^—5 s BP), a lower dilution of the loop circulation water was observed, and the turbidity ratio increased by up to 14 (versus 6) units maximum for 20-s BW filtration (Figure 13d). Consequently, a higher total fouling resistance was observed (Figure 13a,b). The irreversible fouling resistance noticed was 30% higher when BP was used in comparison to BW in the case of V-AE water filtration. The reversible fouling resistance was similar for both conditions tested; thus, BP remained efficient to remove fouling on the membrane side. Results obtained from V-AE water, considered as a low-fouling water, highlighted the role of reverse flow in filtration performance stabilization. A rapid increase in membrane fouling was observed up to complete loss of permeate flow (200 min) with (I) the high reversible fouling resistance and (II) the permeability decreasing during the filtration cycle. This phenomenon appears after 150 min of filtration time when the membrane reverse flow disappears (Figure 13). Reducing the filtration cycle allows limiting the fouling between two BW or BP actions, even when the volume injected is lower. In conclusion, from the results shown, despite the higher irreversible fouling resistance generated on the membrane side, the filtration performance was preserved. Frequent BP actions are beneficial in limiting the concentrate volume produced during similar filtration time without impacting the filtration performance.

Water coming from S-AE was considered as a medium-fouling water. Permeability during its filtration varied between 1200 and 200 L h^−1^ m^−2^ bar^−1^ (Figure 2c). BF operating conditions were studied with a permeate flow of 150 L h^−1^ m^−2^ and continually produced a similar fouling behavior for all experiments (Figure 2c). Similar irreversible fouling resistance over time were observed for each filtration (Figure 14). The BF frequency and duration modification had no impact on irreversible fouling and membrane filtration tendency; however, it helped for increasing the water recovery rate. The application of short and frequent physical cleaning action (BP) on the membrane side allowed us to reach a water recovery rate close to 100% (1 BP 0.5 s/20 min) and increased the turbidity ratio inside the circulation loop up to 10 units. However, the membrane fouling over time was increased in the same way as other conditions. BP was efficient to remove the membrane fouling and recover the permeability value at the beginning of each filtration cycle. The same observation was made for other operating conditions even when the filtration cycle was longer (60 min). Application of a BW or a BP correctly removed fouling particles on the membrane surface and limited the irreversible fouling creation.

### 3.7. Membrane Cleaning and Recovery

The membrane was completely cleaned in place (CIP) before each filtration test. Because a high membrane permeability was noticed and measurement difficulties appeared, a validity range around 20% of the reference permeability was defined to validate the washing efficiency (3,200 L h^−1^ m^−2^ bar^−1^ ± 20%). This limit was respected to start a new filtration experiment. Regarding experimental conditions applied, two complete CIP were sometimes required to recover a good water permeability value.

### 3.8. Membrane Retention Performances

#### 3.8.1. Validation of Seawater Rejection

Several conditions were set for discharging wastewater from exhaust gas cleaning systems, and it must be noted that regulations can change depending on the coastal state in which ships were located. Criteria for seawater discharging of exhaust gas cleaning wastewater was regulated by MEPC 259 (68) resolution [54]. Thus, wastewater can be discharged when the pH value is higher than 6.5, the polyaromatics hydrocarbon (PAH) concentration is lower than 50 µg L^−1^ and turbidity value is not higher than 25 NTU. Permeate samples were analyzed and results show a pH range between 7.5 and 8.5 for all water filtered. The retention rate of suspended solids was close to 100% regardless of the quality of the feed and the operating conditions (permeate turbidity was lower than 6 NTU). Table 2 presents the average turbidity measured in concentrate and in permeate samples and validated the suspended solid elimination after treatment. The PAH concentration was measured in real conditions, and its concentration in permeate samples was lower than the regulation limit. In conclusion, permeate water rejection in seawater is allowed.

#### 3.8.2. Heavy Metals Rejection

A panel of eight metals were chosen due to their presence in plume rejection, including vanadium and nickel, the two main metals involved in the composition of heavy fuel oil used for navigation. The results obtained indicate a retention rate higher than 80% for almost all the metals tested, except for cadmium which was eliminated at 50% (Figure 15). Greater retention was observed with advanced filtration. For instance, vanadium and nickel removal rate increased respectively from 88% to 92% and from 86% to 91% throughout the filtration time (200 min in average) (Figure 15). Similar results show heavy metal removal was obtained with the addition of chemical compounds as an example. Tortora et al. [55] have used surfactants to enhance ultrafiltration in removing zinc, nickel, chromium, and cobalt metal from wastewater with the efficiency of around 88%. BF operations reduced the membrane fouling, which helped to increase the removal rate. Thus, BF influence on heavy metals retention was studied by analyzing permeate samples before and after the water injection. According to Figure 15, the BW applied to the membrane did not impact the retention properties. Heavy metals were adsorbed on the suspended solids surface or were precipitated and perfectly retained by the membrane. According to the literature, MgOH_2_ when used to increase the seawater alkalinity for exhaust gas treatment, helped to precipitate the metal ion by formation of metal hydroxides throughout the time [24,56,57]. In conclusion, the usage of the membrane process with high absorptive and nontoxic suspended solids can replace the addition of chemical compounds for the elimination of heavy metals from wastewater.

### 3.9. Performances Validations on Onboarded Membrane Separation Units

Membrane filtration is used on maritime vessels, such as the CC-LOUIS BLERIOT (CC-LB), to treat process water and reduce the suspended solids concentration. Units installed on maritime vessels used SiC membranes, and they treat around 8 m^3^ h^−1^ of process water when the scrubber unit is running in closed loop. Approximately 5% of this water flow is continuously eliminated as concentrate water and goes to the residue tank. The remaining 95%, considered as permeate water, is either discharged to seawater, reinjected into the unit during BW action, or returned back to the process tank. BW actions are usually performed every 20 min on units which correspond to a filtration interval of 80 min on each membrane module (1 unit is composed of 2 lines in series with 2 modules k99 in parallel). Because BW water is sent to a process tank, the average residue volume produced in 1 h of filtration is around 400 L. This value is quite important due to the low residue tank volume available on marine vessels. For instance, on a CC-LB container ship, the residue volume is 85 m^3^ which allows the CL to run for only eight days without issues. Due to the time spent in European SECA, around 15 to 20 days, a function of port availability, the eight-day CL limitations are critical values for the ship navigation. Indeed, in SECAs, sulfur concentration allowed in ships’ plume rejection is 0.1%; thus once the residue tank is full, the ship must switch from high sulfur fuel to diesel or low sulfur fuel which is more expensive (around 200$ t^−1^ difference). Currently, to limit the membrane fouling and residue volume, the membrane unit is operated with low permeate flow, lower than 29 L h^−1^ m^−2^ with only one filtration line in service, which reduces the membrane operation flexibility.

Membrane feedwaters were analyzed, and a value of turbidity at 170 NTU was found, with suspended solids at 1.2 g L^−1^ and dry matter at 55 g L^−1^, which were in the range of high-fouling fluid. Thus, the filtration performance of CC-LB can be compared to KERG or V-ME results obtained from semi-industrial scale experiments even though the CC-LB membrane feedwater was pretreated (coagulation, precipitation, and hydro-cyclone) before the membrane separation. For the reasons of low residue tank volume and high-fouling effluent, the ship CC-LB was chosen to validate the scale results. The onboard membrane filtration unit was studied under different scrubber running conditions, in OL with constant process water quality, in CL during the navigation with both engines started, and in CL during port stay with scrubber 1 (ME) out of service. Permeate flow was increased from 29 to 63 L h^−1^ m^−2^, a value recommended from scale tests. To get as close as possible to semi-industrial scale experiments and decrease the residue volume, no concentrate water was eliminated continuously from the unit. The filtration loop was only purified during BW action. Results obtained confirm the conclusion of semi-industrial scale tests. A low membrane fouling throughout the filtration time was observed no matter the scrubber operation (OL/CL). Permeability measured was between 70 and 130 L h^−1^ m^−2^ bar^−1^ and the TMP values were lower than 0.7 bar, the maximum TMP value for making a CIP.

To illustrate the membrane performance, Figure 16 shows the filtration tendency in terms of permeability, TMP, and fouling resistance for the onboard membrane unit when scrubber 2 was run in CL in hoteling. The unit runs with a permeate flow of 63 L h^−1^ m^−2^ bar^−1^ for 17 h with only a slow increase in membrane fouling over time thanks to the BW application. After 13 h of filtration, the maximum BW TMP (0.65 bar) was reached many times before the end of the filtration step delay (BW initiated with timers) and resulted in an irreversible resistance stabilization. Nevertheless, the maximum CIP TMP (0.7 bar) was not reached. Additionally, reducing the permeate flow allows for decreasing the membrane filtration constraints; consequently, the treatment of process water was maintained for extra hours, giving time to quit controlled areas and switch to OL for instance.

A similar permeate flow is applied to both units (semi-industrial and onboarded unit) for high-fouling water filtration. In both cases a permeability stabilization over time is observed and maintained thanks to the BW actions. Usage of shorter filtration cycles helps the membrane to maintain a lower TMP value in cases where more polluted water needed to be treated. Moreover, in comparison with the semi-industrial scale experiment, onboarded feedwater is pretreated, which decreases the fouling particle concentration in feedwater. However, due to the longer filtration cycle required by the unit configuration (two BW in 40 min and then no BW for 60 min) the recovery rate was higher. A global recovery rate of 90% was obtained and led to a reduction of the residue volume produced. The suspended solids in the filtration were significant, approximately 16 g L^−1^. Permeate quality was always within discharge criteria range with an alkaline pH (8.7), a turbidity of 10.5 NTU (<25 NTU), and a PAH concentration lower than 50 µg L^−1^. Membrane-retention properties were not influenced by the feedwater quality, and the permeate produced was still compliant with discharge regulations. Heavy metal removal efficiency was also studied. The analysis highlights, first their higher concentration in real water than in the process water received for scale tests (for example, between 24 and 130 mg L^−1^ for vanadium) and the membrane was able to retain 94, 96, and 99%, respectively, of nickel, vanadium, and aluminum metal ions having the highest concentration in treated suspension (>5 mg L^−1^). Experiments done on CC-LB validate the scale filtration results and highlight a good membrane operational flexibility for the crew. Results show that the membrane was able to maintain a higher permeate flow if, for instance, it was needed to drain the process tank with the limitation of residue volume production. Additionally, the limitation of eight closed loop days can be increased as a function of the process water quality and filtration condition.

## 4. Conclusions

In this paper, SiC membrane filtration was studied as an alternative for treating scrubber wastewater. This is the first time that filtration and separation performance are reported in the literature for maritime scrubber’s water treatment applications. A large range of water quality was filtered, and the influence of operating conditions such as permeate flux, BF frequency, and duration types were studied for each effluent categorized as high- and low-fouling water. The results obtained highlight the following points: (I) increase in permeate flow and the filtration step led to an increase in the irreversible resistance; (II) reduction of BW duration until a certain value did not impact membrane performance in the long term because the fouling layer cake was correctly removed; and (III) usage of BP instead of BW helps to maintain the filtration performance by reducing the concentrate volume eliminated. Permeate flow values and BF conditions were prescribed for the treatment of scrubber water from a semi-industrial study and validated under real operating applications on an onboarded vessel filtration unit. Thus, for the first time, flexible operating conditions applicable to the entire fleet are defined. For high-fouling fluid properties, a maximum permeate flow of 65 L h^−1^ m^−2^ is applied with frequent and rapid BF action as BP. The BP action helping to reduce the membrane fouling through the time for longer filtration duration. Under these conditions, a recovery rate of 96% can be obtained on an industrial scale, which largely reduces the residue volume production in comparison to current conditions where the recovery rate is around 90%. According to semi-industrial scale experiments, operating conditions are also defined for low- and medium-fouling fluid properties. For lower-fouling fluid, higher permeate flow can be applied up to 150 L h^−1^ m^−2^ with BW initiated every 40 or 60 min helping to quickly drain process tanks. Cases of medium-fouling water 150 L h^−1^ m^−2^ appear as the best permeate flux. Coupling this permeates flux with a short BP action every 20 min can greatly increase the water recovery, up to 99%, which reduces the concentrate volume. In terms of permeate water quality, the analysis performed allows its rejection to the natural environment. Discharge criteria were validated, the membrane particle retention was close to 99%, and heavy metal removal higher than 80% from the beginning of the filtration step for each operating condition.

Usage of the membrane process coupled with the exhaust gas cleaning system is useful to uphold environmental regulations (air and water). Studying and understanding the membrane filtration tendency is important and initially described in this paper. The tests carried out with these high values of permeate flow confirm the results obtained with high water-recovery rates, and without major consequences for the membrane installation, giving greater freedom of action. Onboard, the operational constraints are strong with four tanks to manage simultaneously. Individually, the two process tanks must not be too empty to ensure the very high recirculation flow rates of the scrubbers (400 and 1400 m^3^ h^−1^ respectively for ME and AE scrubber), nor too full, which would force them to work in closed loop and generate residues and permeate. The volume in the residue tank must not increase too quickly, as this would lead to frequent emptying at the port or the use of more expensive fuel. The permeate tank must not fill up too quickly at the pier, as emptying is prohibited. This freedom of action also eases the stress, as only 40 people manage a vessel such as the CC LOUIS BLERIOT.

## Figures and Tables

**Figure 1 membranes-12-00968-f001:**
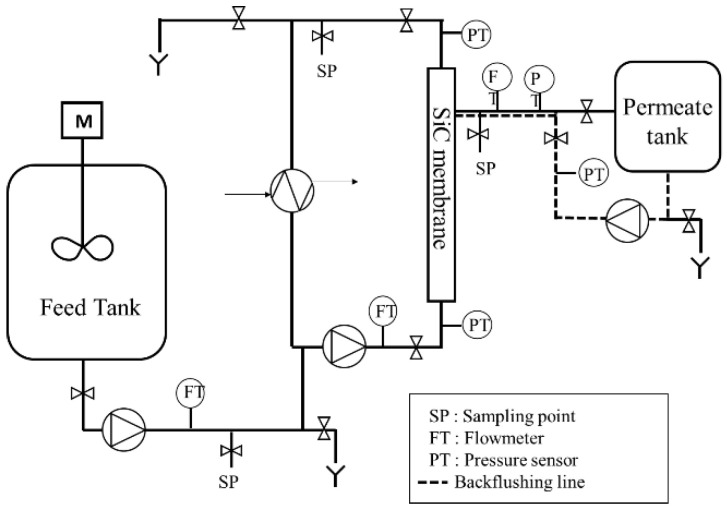
Simplified pilot plant scheme.

**Figure 2 membranes-12-00968-f002:**
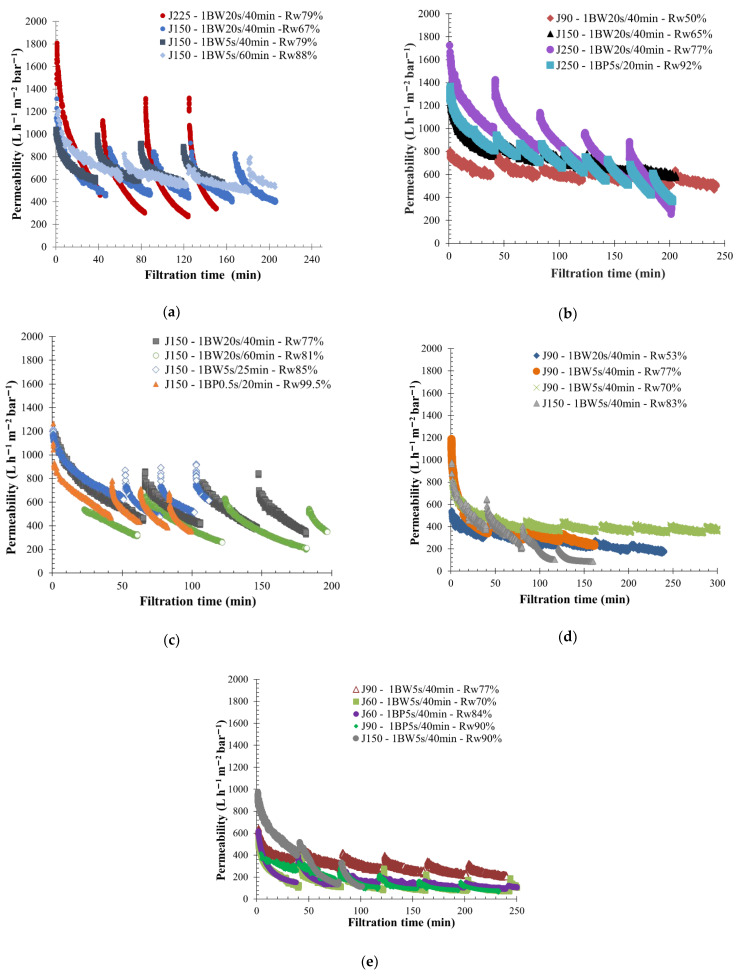
Overview of membrane behavior for all fluid quality tested by variation of the permeability throughout the filtration time. T = 20 °C; SiC membrane 0.33 m^2^. (**a**) APL-SINGAPURA-ME; (**b**) APL VANDA AE; (**c**) APL-SINGAPURA AE; (**d**) APL-VANDA ME; (**e**) CC-KERGUELEN. Jxx is permeate flux at the xx value in L h^−1^ m^−2^; 1BW yy s/zz min where yy is the duration and zz the interval of filtration between two BF (BW or BP as it is mentioned in the legend); and Rw is the total recovery rate applied to the unit.

**Figure 3 membranes-12-00968-f003:**
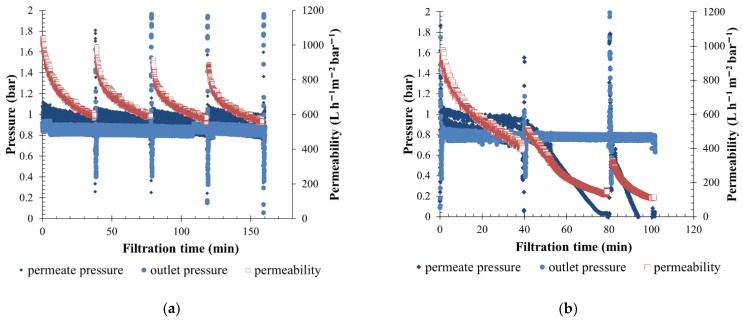
Variation of permeate pressure, outlet pressure and permeability value measured throughout the filtration time for (**a**) low-fouling water type, APL-SINGAPURA ME —J150—Rw = 79%; and (**b**) high-fouling water, CC-KERGUELEN J150—Rw = 90%.1 BW 5 s/40 min; T = 20 °C; SiC membrane 0.33 m^2^.

**Figure 4 membranes-12-00968-f004:**
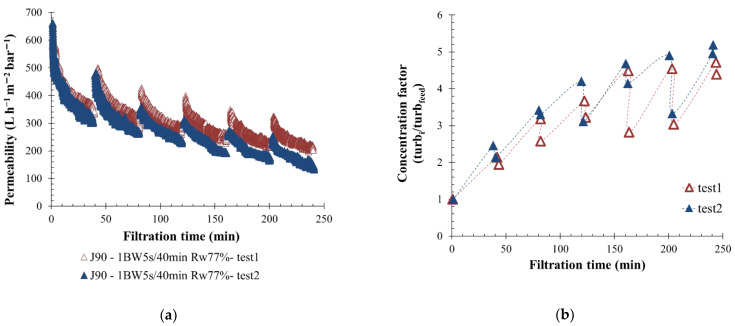
Repeatability experiment with a permeate flux of 90 L h^−1^ m^−2^. Variation of permeability at 20 °C (**a**) and turbidity concentration factor from the feed value (**b**) throughout the time with backwash water injected for 5 s every 40 min. T = 20 °C, feedwater from CC-KERGUELEN; SiC membrane 0.33 m^2^. Jxx is permeate flux at the xx value in L h^−1^ m^−2^; 1BW yy s/zz min where yy is the duration and zz the interval of filtration between two BF; and Rw is the total recovery rate applied on the unit.

**Figure 5 membranes-12-00968-f005:**
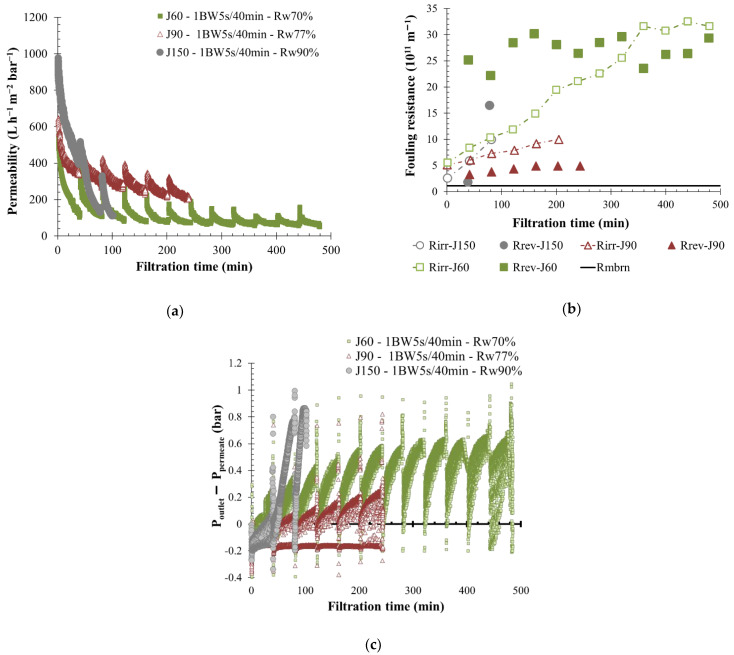
Influence of permeate flux on membrane filtration behavior by variation of permeability at 20 °C (**a**), fouling resistance (**b**), and the pressure ration, difference between membrane outlet, and permeate side pressure (**c**) throughout the time with backwash water injected for 5 s every 40 min. A flux of 150 L h^−1^ m^−2^ (grey round), 90 L h^−1^ m^−2^ (red triangle) and 60 L h^−1^ m^−2^ (green square) are applied to the membrane. T = 20 °C, feedwater from CC-KERGUELEN; SiC membrane 0.33 m^2^. Jxx is permeate flux at the xx value in L h^−1^ m^−2^; 1BW yy s/zz min where yy is the duration and zz the interval of filtration between two BF; and R_w_ is the total recovery rate applied on the unit; R_irr_ is irreversible resistance; R_rev_ is reversible resistance and R_mbrn_ is the membrane resistance.

**Figure 6 membranes-12-00968-f006:**
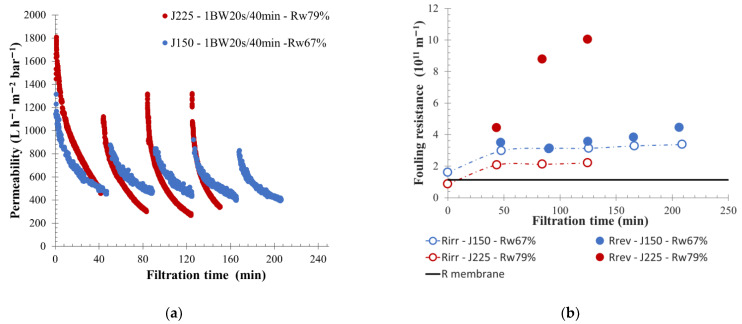
Influence of permeate flux on membrane filtration behavior by variation of permeability at 20 °C (**a**) and fouling resistance (**b**) throughout the time with backwash water injected for 20 s every 40 min. Flux of 225 L h^−1^ m^−2^ (round), 150 L h^−1^ m^−2^ (triangle) is applied to the membrane. T = 20 °C, feedwater from APL-SINGAPURA-ME; SiC membrane 0.33 m^2^. Jxx is permeate flux at the xx value in L h^−1^ m^−2^; 1BW yy s/zz min where yy is the duration and zz the interval of filtration between two BF; and R_w_ is the total recovery rate applied on the unit; R_irr_ is irreversible resistance; R_rev_ is reversible resistance and R_membrane_ is the membrane resistance.

**Figure 7 membranes-12-00968-f007:**
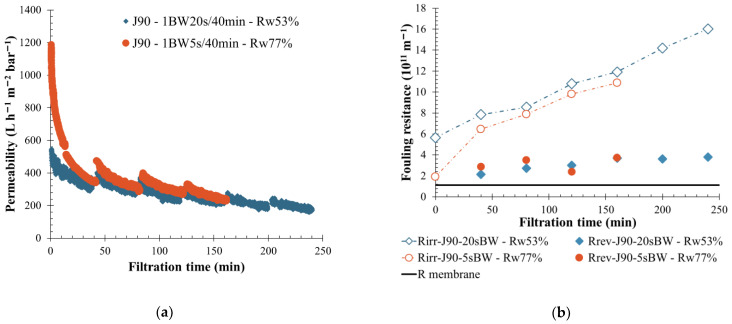
Influence of BW duration on membrane filtration behavior by variation of permeability at 20 °C (**a**) and fouling resistance (**b**) throughout the time to permeate flux of 90 L h^−1^ m^−2^, a BW interval of 40 min and a BW duration of 5 s (orange round) and 20 s (blue diamond) are applied to the membrane. T = 20 °C, feedwater from APL-VANDA-ME; SiC membrane 0.33 m^2^. Jxx is permeate flux at the xx value in L h^−1^ m^−2^; 1BW yy s/zz min where yy is the duration and zz the interval of filtration between two BF; and Rw is the total recovery rate applied on the unit; R_irr_ is irreversible resistance; R_rev_ is reversible resistance and R_membrane_ is the membrane resistance.

**Figure 8 membranes-12-00968-f008:**
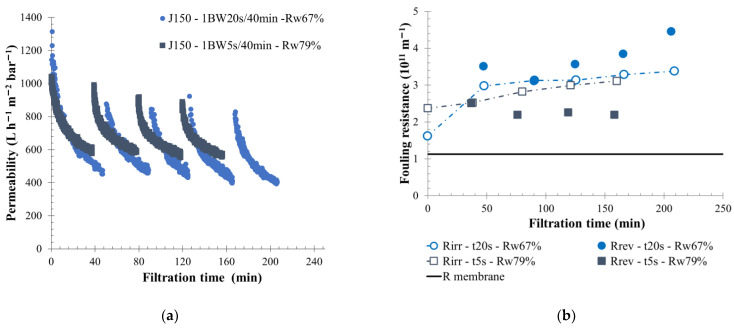
Influence of BW duration on membrane filtration behavior by variation of permeability at 20 °C (**a**) and fouling resistance (**b**) throughout the time with permeate flux of 150 L h^−1^ m^−2^, a BW interval of 40 min and a BW duration of 5 s (squared) and 20 s (round) are applied to the membrane. T = 20 °C, feedwater from APL- SINGAPURA-ME; SiC membrane 0.33 m^2^. Jxx is permeate flux at the xx value in L h^−1^ m^−2^; 1BW yy s/zz min where yy is the duration and zz the interval of filtration between two BW; and Rw is the total recovery rate applied on the unit; R_irr_ is irreversible resistance; R_rev_ is reversible resistance and R_membrane_ is the membrane resistance.

**Figure 9 membranes-12-00968-f009:**
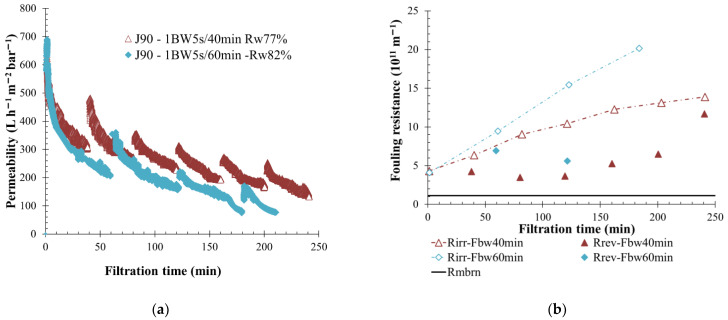
Influence of BW frequency on membrane filtration behavior by variation of permeability at 20 °C (**a**) and fouling resistance (**b**) throughout the time with permeate flux of 90 L h^−1^ m^−2^ and a BW duration of 5 s, every 40 min (red triangle) and every 60 min (blue diamond) are applied to the membrane. T = 20 °C, feedwater from CC-KERGUELEN; SiC membrane 0.33 m^2^. Jxx is permeate flux at the xx value in L h^−1^ m^−2^; 1BW yy s/zz min where yy is the duration and zz the interval of filtration between two BF; and R_w_ is the total recovery rate applied on the unit; R_irr_ is irreversible resistance; R_rev_ is reversible resistance and R_mbrn_ is the membrane resistance.

**Figure 10 membranes-12-00968-f010:**
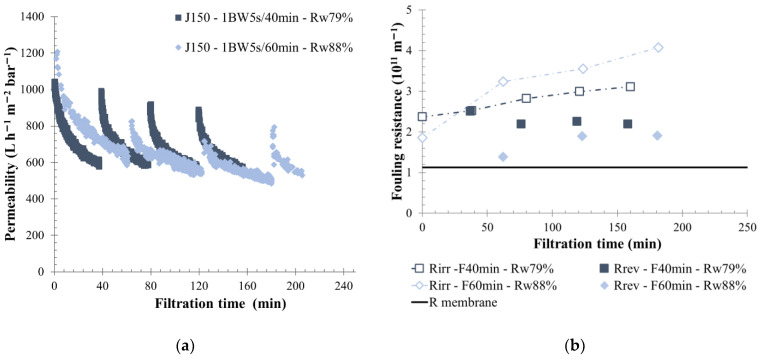
Influence of BW frequency on membrane filtration behavior by variation of permeability at 20 °C (**a**) and fouling resistance (**b**) throughout the time with permeate flux of 150 L h^−1^ m^−2^ and a BW duration of 5 s, every 40 min (squared) and every 60 min (diamond) are applied to the membrane. T = 20 °C, feedwater from APL-SINGAPURA ME; SiC membrane 0.33 m^2^. Jxx is permeate flux at the xx value in L h^−1^ m^−2^; 1BW yy s/zz min where yy is the duration and zz the interval of filtration between two BF; and R_w_ is the total recovery rate applied to the unit; R_irr_ is irreversible resistance; R_rev_ is reversible resistance and R_membrane_ is the membrane resistance.

**Figure 11 membranes-12-00968-f011:**
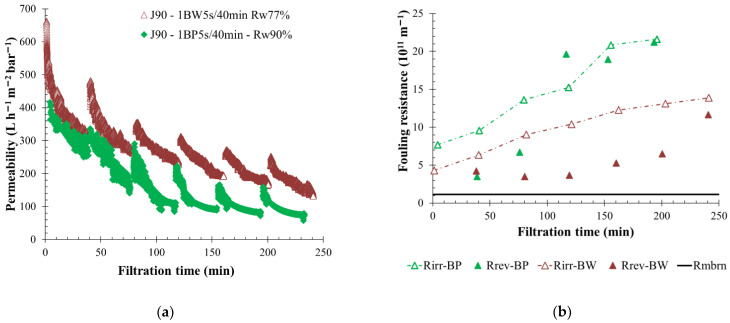
Influence of backflush mode (BP-BW) on membrane filtration behavior by variation of permeability at 20 °C (**a**); fouling resistance (**b**) and the concentration factor over time with pressure water injected for 5 s every 40 min and a permeate flux of 90 L h^−1^ m^−2^. Filtration is made with BW action (white triangle) and BP action (grey triangle). T = 20 °C, feedwater from CC-KERGUELEN; SiC membrane 0.33 m^2^. Jxx is permeate flux at the xx value in L h^−1^ m^−2^; 1BW yy s/zz min where yy is the duration and zz the interval of filtration between two BF; and Rw is the total recovery rate applied on the unit; R_irr_ is irreversible resistance; R_rev_ is reversible resistance and R_mbrn_ is the membrane resistance.

**Figure 12 membranes-12-00968-f012:**
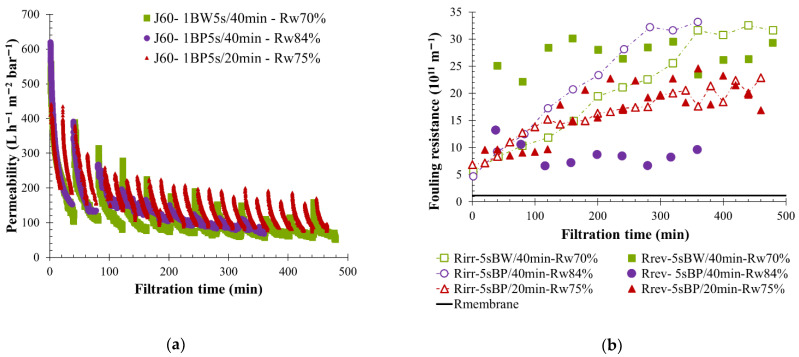
Influence of backflush conditions on membrane filtration behavior by variation of permeability at 20 °C (**a**); fouling resistance (**b**) throughout the time with pressured water injected during 5 s and a permeate flux of 90 L h^−1^ m^−2^. The filtration conditions as BF mode, frequency and volume injected are, respectively, BW, 40 min, 5.5 L (squared); BP, 40 min, 1.7 L (round) and BP, 20 min, 1.7 L (triangle). T = 20 °C, feedwater from CC-KERGUELEN; SiC membrane 0.33 m^2^. Jxx is permeate flux at the xx value in L h^−1^ m^−2^; 1BW yy s/zz min where yy is the duration and zz the interval of filtration between two BF; and Rw is the total recovery rate applied on the unit; R_irr_ is irreversible resistance; R_rev_ is reversible resistance and R_membrane_ is the membrane resistance.

**Figure 13 membranes-12-00968-f013:**
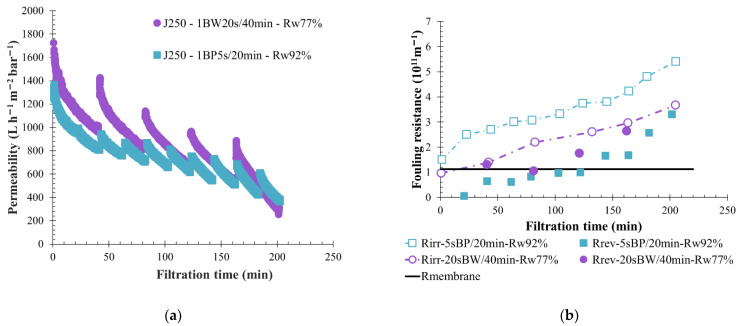
Influence of backflush mode (BP-BW) on membrane filtration behavior by variation of permeability at 20 °C (**a**); fouling resistance (**b**), the pressure ratio (**c**) and the turbidity concentration ration (**d**) throughout the time for a permeate flux of 250 L h^−1^ m^−2^ and a 1 BW 20 s/40 min (purple round) and 1 BP 5/20 min (blue squared). T = 20 °C, feedwater from APL-VANDA-AE; SiC membrane 0.33 m^2^. Jxx is permeate flux at the xx value in L h^−1^ m^−2^; 1BW yy s/zz min where yy is the duration and zz the interval of filtration between two BF; and Rw is the total recovery rate applied on the unit; R_irr_ is irreversible resistance; R_rev_ is reversible resistance and R_membrane_ is the membrane resistance, turb_i_ is the turbidity of the sample, turb_feed_ is the turbidity of the feed water.

**Figure 14 membranes-12-00968-f014:**
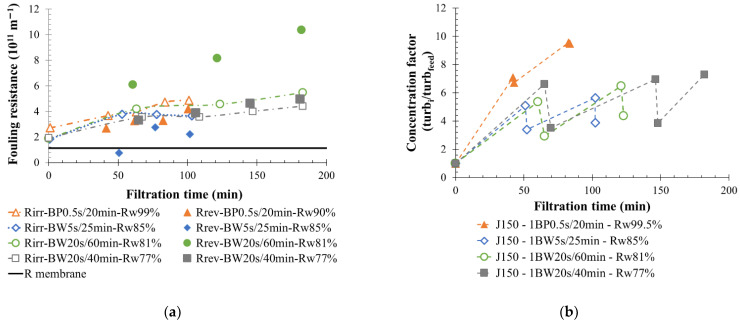
Influence of backflush mode (BP-BW) on membrane filtration behavior by variation fouling resistance (**a**), and the turbidity concentration ration (**b**) throughout the time for a permeate flux of 150 L h^−1^ m^−2^ and a 1 BP 0.5 s/20 min (triangle); 1 BW 20 s/40 min (squared); 1 BW 20 s/60 min (round); 1 BW 5 s/25 min (diamond). T = 20 °C, feedwater from APL-SINGAPURA-AE; SiC membrane 0.33 m^2^. Jxx is permeate flux at the xx value in L h^−1^ m^−2^; 1BW (or BP) yy s/zz min where yy is the duration and zz the interval of filtration between two BW (or BP); and R_w_ is the total recovery rate applied to the unit; R_irr_ is irreversible resistance; R_rev_ is reversible resistance and R_membrane_ is the membrane resistance.

**Figure 15 membranes-12-00968-f015:**
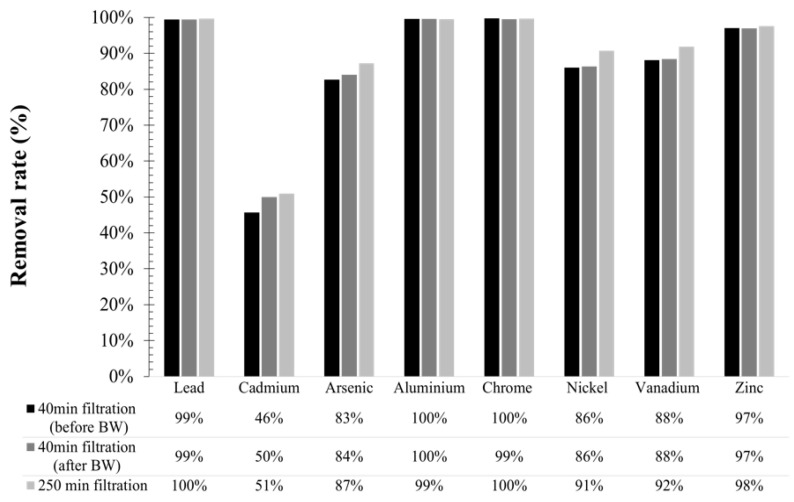
Heavy metal retention and impact of BW action for high-fouling water.

**Figure 16 membranes-12-00968-f016:**
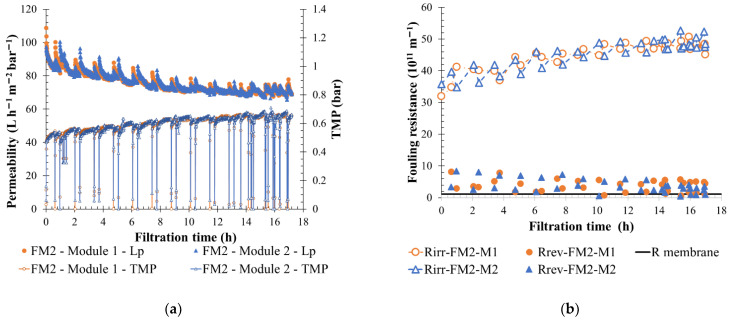
Variation of permeability at 20 °C and TMP (**a**), fouling resistance (**b**) throughout the filtration time with BW initiate every 20 min on the entire membrane filtration unit −1/80 min on one filtration module, a permeate flux of 63 L h^−1^ m^−2^. T = 20 °C, feedwater from CC-LOUIS BLERIOT process tank 2 after pre-treatment steps; 1 line of 2 module k99 in parallel SiC membrane 32.67 m^2^/module. FMx is the line considered, Mx is the membrane, Lp is the permeability value, TMP the transmembrane pressure and Rw is the total recovery rate applied on the unit; R_irr_ is irreversible resistance; R_rev_ is reversible resistance and R_membrane_ is the membrane resistance.

**Table 1 membranes-12-00968-t001:** Physical and chemical characteristics of effluents treated (TSS is made from NF EN 872 protocol and dry matter from NF EN 12880).

Vessel	Turbidity (NTU)	pH (-)	Conductivity (mS cm^−1^)	TSS * (g L^−1^)	Dry Matters (g L^−1^)
APL SINGAPURA—ME	85	8.5	47.7	0.47	36
APL VANDA—AE	306	6.0	55.8	0.59	67
APL SINGAPURA—AE	105	8.2	65.7	0.61	88
APL-VANDA—ME	553	8.8	59.4	1.15	87
CMA CGM KERGUELEN	214	7.6	81.9	0.55	127

* TSS, total suspended solid.

**Table 2 membranes-12-00968-t002:** Turbidity range in concentrate side and permeate side during experiments.

	Range of Loop Turbidity Value Throughout the Filtration Experiment (NTU)	Average Permeate Turbidity Value (NTU)
Effluent 1	≈140–2200	0.7
Effluent 2	≈306–4600	0.3
Effluent 3	≈23–259	0.3
Effluent 4	≈553–3300	2.2
Effluent 5	≈180–1300	6.0

## Data Availability

In this section, please provide details regarding where data supporting reported results can be found, including links to publicly archived datasets analyzed or generated during the study.

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
