# Peer review of "Membrane Separation Used as Treatment of Alkaline Wastewater from a Maritime Scrubber Unit"

_membranes, 2022, doi:10.3390/membranes12100968_

Round 1

Reviewer 1 Report

This manuscript presents the performance of ultrafiltration membrane in the maritime scrubber unit for wastewater filtration. 

1. Page 1, line 43: For comma, "." instead of "," should be used.

2. Page 2, line 57: The word "gaz" should be changed to "gas".

3. The authors may consider to include more references that are more directly relevant to membrane technology in scrubber unit. 

4. Page 3, lines 101-109: For the applications of SiC membranes, are the processes for lab-scale research or industrial-scale treatments?

5. Page 3, lines 115-116: May the authors explain how high permeability reduces membrane fouling?

6. Section 2: The description of pilot and experimental procedure are very long. It may be shortened as readers of this work have relevant background.

7. Page 8, lines 272-274: Is "reverse flux" noticed for subsequent filtration cycles? 

8. Though SiC membrane shows a pure water permability above 3000 LMH/bar, the actual permeate flow is below 100 LMH/bar. Its value of "high flux" is greatly reduced. Therefore, it may not matter whether PWP is 3000 or 1000 LMH. Feed water quality as well well the membrane chemistry determine the fouling behavior. 

Author Response

This manuscript presents the performance of ultrafiltration membrane in the maritime scrubber unit for wastewater filtration. 

  1. Page 1, line 43: For comma, "." instead of "," should be used.

Thanks for your remark, the mistake has been corrected.

  1. Page 2, line 57: The word "gaz" should be changed to "gas".

Thanks for your remark, the mistake has been corrected.

  1. The authors may consider to include more references that are more directly relevant to membrane technology in scrubber unit. 

Thank you very much for your comment, currently any paper available in the literature related to the usage of membrane or performances for maritime scrubber wastewater. However, we have taken into consideration your remark and added 2 sentences about membrane usage for onshore desulfurization plant in lab-scale application, but feed waters weren’t the same as marine scrubber water.

  1. Page 3, lines 101-109: For the applications of SiC membranes, are the processes for lab-scale research or industrial-scale treatments?

Currently, SiC membranes are used in industrial scale for some application in diverse fields as mentioned in manuscript and in article written from ERAY et al 2021. However, somes articles presented in the paper are lab scale investigation to increase the knowledge of these membranes’ materials.

  1. Page 3, lines 115-116: May the authors explain how high permeability reduces membrane fouling?

Thanks for the very interesting question. We have modified our sentence and added some sentences. The high permeability of SiC membranes can be benefic for water treatment in comparison to other membrane material. For a similar fouling rate, the flux applied can be higher which helps to maintain the production value throughout the time. This question was also asked by the other reviewer so more explanation have been added to the manuscript helping for comprehension with the study of Hofs et al. to justify the statement.

  1. Section 2: The description of pilot and experimental procedure are very long. It may be shortened as readers of this work have relevant background.

Thanks for your remark, we have taken it into account and corrected it in the article by removing no relevant information.

  1. Page 8, lines 272-274: Is "reverse flux" noticed for subsequent filtration cycles? 

Thanks for your question, reverse flux is rapidly explained in the article, reverse flux is mainly observed in the beginning of the filtration and after each backflush action due to the removal of fouling layers. However, according to fouling properties of the fluid, more or less reverse flux is present on membrane side (near the outlet of the membrane). As an example, a low fouling water type, the fouling resistance was always low and the entire membrane surfaces were not fouled as the same way, coupled with the low tortuosity of the membrane surface, the water can easily pass through the membrane from permeate to feed. In contrary, for a high-fouling fluid, the membrane fouling was rapid and important, the resistance in membrane surface is sufficient to limit the reverse flux.

  1. Though SiC membrane shows a pure water permeability above 3000 LMH/bar, the actual permeate flow is below 100 LMH/bar. Its value of "high flux" is greatly reduced. Therefore, it may not matter whether PWP is 3000 or 1000 LMH. Feed water quality as well the membrane chemistry determine the fouling behavior. 

Your remark is very interesting, thanks you very much to highlight this point. Indeed, the fouling behavior depends on the feed water quality and the membrane properties define the fouling behavior. However, if the initial permeability is high, higher flux can be obtained for the same fouling rate (addition of membrane resistance and membrane fouling). Additionally, the low tortuosity of the membrane characteristic to SiC membrane from the support to the skin induces a high permeability. It helped to reduce energies needed for backflushing and enhance these actions. Because permeability is high, water can easily pass through the membrane and efficiency remove the fouling. 

Reviewer 2 Report

This manuscript reports a work on the purification of alkaline wastewater from ships via a commercial SiC membrane. The work is interesting and has important practical implications. In the current version, the author presents a lot of data, however, the structure and theme of the article are not prominent enough, the overall presenting is not concise enough, and reading comprehension is difficult. Therefore, I recommend accepting it after a major revision and language polishing.

1.     Overall, the language should be improved. The grammar should be thoroughly checked and corrected. For example:

In the abstract:

“on this paper”, “A range of operating parameters to improve (i) the water recovery rate, (ii) the filtration duration and (iii) the permeate quality was obtained for several feed water and applied to industrial scale units.” A range of operating parameters … were obtained for ?

“Similar results were obtained in scale and industrial units.Does the author mean lab scale?

In the abstract, should provide the full name of PAH”.

2.     Line 150-152: “On the membrane filtration skid used for filtration, downstream of the feed tank, a feed pump was used to fix the inlet pressure at 1.5 bar.” What is the membrane filtration skid used for filtration?

3.   Lines 171 and 169: The authors use BW and BP in lines 162-163 to indicate backwash and backpulse. Abbreviations should then be used in the following contexts. There are many similar writing typos.

“*BF: backflush actions, refer to backwash or backpulse triggered on membrane side”

In table 2, the author uses another term backflush to indicate the BW and BP, seems do not differentiate these two terms. Should be careful of defining terms.

In line 399 and line 507, Backwash (BW) was denoted again.

4.     Line 415, please check the expression of value “around 3 1011 m1

5. The format of figrues can be improved to be clear if presenting in black and white. 

Author Response

R2:

This manuscript reports a work on the purification of alkaline wastewater from ships via a commercial SiC membrane. The work is interesting and has important practical implications. In the current version, the author presents a lot of data, however, the structure and theme of the article are not prominent enough, the overall presenting is not concise enough, and reading comprehension is difficult. Therefore, I recommend accepting it after a major revision and language polishing.

Thanks for accepting to review our manuscript and thank you for your interesting comments. Your comment regarding the reading comprehension and the structure have been noted. Consequently, the introduction has been modified by adding information about the fluid classification and the industrial needs regarding the application. We hope that this will help the ready to better understand the importance of the study and the order in which the results are presented. Moreover, the entire manuscript has been revised and language corrected by a native scientific English speaker.

  1. Overall, the language should be improved. The grammar should be thoroughly checked and corrected. For example:

In the abstract:

“on this paper”, “A range of operating parameters to improve (i) the water recovery rate, (ii) the filtration duration and (iii) the permeate quality was obtained for several feed water and applied to industrial scale units.” A range of operating parameters … were obtained for ?

Thanks for this remark, mistakes have been corrected in manuscript.

“Similar results were obtained in scale and industrial units.” Does the author mean lab scale?

Thanks for highlighted this point, in this case lab-scale with a half industrial unit has been used for filtration experiment with the five effluents presented. From results obtained in lab-scale, operating conditions have been defined for each effluent category and then applied to an industrial scale unit, running with 198 membranes at the same time. During industrial tests, the feed water was highly concentrated in pollutant, so we were in high fouling water cases and applied operating parameter in consequence (J=60Lh-1m-2). We have modified the sentences.

In the abstract, should provide the full name of “PAH”.

Thanks for your remarks, all mistakes have been corrected

  1. Line 150-152: “On the membrane filtration skid used for filtration, downstream of the feed tank, a feed pump was used to fix the inlet pressure at 1.5 bar.” What is the membrane filtration skid used for filtration?

Thanks for your question the membrane filtration skid used is described and schematized in figure 1. We used a cross flow circulation mode in a multi-tubular SiC membranes. The scheme refers to the half industrial unit. However, on board unit work with the same way but more membranes are installed in parallel. Any of permeate nor concentrated return to the feed tank. However, other reviewers ask to reduce the material and method by removing general information, thus we simplified this part.

  1.  Lines 171 and 169: The authors use BW and BP in lines 162-163 to indicate backwash and backpulse. Abbreviations should then be used in the following contexts. There are many similar writing typos.

“*BF: backflush actions, refer to backwash or backpulse triggered on membrane side”

Thanks for your remarks, writing typos regarding abbreviation have been corrected in the manuscript. Regarding your comment, backflush is a general term used for both action with no distinction. We employed this term when we want to discuss of the water injection throughout the membrane and not specifically a BW or a BP.

  1.   In table 2, the author uses another term backflush to indicate the BW and BP, seems do not differentiate these two terms. Should be careful of defining terms.

Thank you for highlighting this point, regarding Table 2, we had not differentiated both actions because there are never used at the same time, we wanted to highlight the reverse flow action in general and not specifically BW or BP. Questions on table 2 appeared several times in the reviewer comment, thus we changed the format of the table to improve the comprehension by integrating it in text.  

In line 399 and line 507, Backwash (BW) was denoted again.
Thanks for reminder, all mistakes have been corrected

  1. Line 415, please check the expression of value “around 3 1011 m−1”

Thank you for your advising. This value has been checked and refers to the figure 7 below. To remove any ambiguity or reader question we replace the expression “around 3 1011 m—1” by “below 4 1011 m-1

  1. The format of figrues can be improved to be clear if presenting in black and white. 

Thanks for your remark, figures have been modified and presenting in color for better understanding.

Reviewer 3 Report

The manuscript prepared by Drouin et al. presents a work to illustrate the applicability of the membrane system in treating alkaline wastewater from the maritime scrubber unit. The main objective of this work focused on evaluating a range of operating parameters to improve the treatment performance of the membrane separation process. This work provides valuable evidence on the efficiency of using membrane systems in treating “real” wastewater in practice. Also, the manuscript is well written with comprehensive data to support its conclusion. It is a good work. I would recommend acceptance after addressing comments below.

Major comments:

1.      I would like to suggest the authors revise the manuscript thoroughly by adding proper citations to the statements that do not come from you. For example, line 31 – 32, “For instance……compared to 2021”. This is just one typical example, there are other statements also need proper citations.

2.      I suggest the author pay more attention to writing, excluding any typos in the manuscript. For example, line 57: “gaz (gas)”, line 179: “when evaluating evaluate”. Please review the manuscript thoroughly to exclude such mistakes in the following version.

3.      Line 115 – 117: “In terms of membrane performance …… and effluent storage constraints”. This statement does not come with citations, I would assume it was from the authors. However, from the perspective of the membrane process, high permeability would enhance, rather than reduce, membrane fouling due to concentration polarization at the interface between the membrane and the solution. Thus, can the authors explain this statement or provide a proper citation to support this?

4.      For table 2, the operating conditions applied on SiC membranes are not applicable to all water sources (five different effluents), I am confused about the presence of such data here.

5.      For Figure 2(a-e), it is very difficult to compare each figure because both the y and x axis are different. I would suggest the author replot the figure 2 to clearly indicate the difference among effluents. Also, please add distinct colors for all figures in the manuscript.

6.      Line 333 - 334: “Application of a permeate flow…as it is shown in Figure 5”. In RO, a high permeability would result in severe concentration polarization that leads to more membrane fouling. I am curious why you observed differently. Can you explain in more detail why you observed this?

7.      I am curious about what types of fouling that caused the membrane to fail in your experiments. I am aware that you have five effluents with different fouling potential. Can you indicate specifically what kind of fouling (organic, scaling or biofouling) is for each effluent? Throughout the manuscript, I don’t see the authors explain the fouling.

8.      Also, I am curious what are the irreversible and reversible fouling, do you have SEM images? The type of fouling needs to be identified because, in practice, you might need different strategies to handle different fouling. 

Author Response

R3

The manuscript prepared by Drouin et al. presents a work to illustrate the applicability of the membrane system in treating alkaline wastewater from the maritime scrubber unit. The main objective of this work focused on evaluating a range of operating parameters to improve the treatment performance of the membrane separation process. This work provides valuable evidence on the efficiency of using membrane systems in treating “real” wastewater in practice. Also, the manuscript is well written with comprehensive data to support its conclusion. It is a good work. I would recommend acceptance after addressing comments below.

Major comments:

  1. I would like to suggest the authors revise the manuscript thoroughly by adding proper citations to the statements that do not come from you. For example, line 31 – 32, “For instance……compared to 2021”. This is just one typical example, there are other statements also need proper citations.

Thanks for your remark, manuscript have been revised. Regarding the sentence highlights here line 31-32, it comes from Suez Canal Data extracted at different moment of the study and translate in the article. This example as the majority of these informations are common to all maritime transport actors. Regarding other statement, some of them come from our feedback on the subject and it can’t be justified from literature as example the lines 106-108 “treat the scrubber water mainly composed of natural salty water, hydrocarbons, heavy metal, particulate matter, and unburned fuel residue”.

  1. I suggest the author pay more attention to writing, excluding any typos in the manuscript. For example, line 57: “gaz(gas)”, line 179: “when evaluating evaluate”. Please review the manuscript thoroughly to exclude such mistakes in the following version.

Thanks for your remark, the manuscript has been reviewed and typos errors have been corrected.

  1. Line 115 – 117: “In terms of membrane performance …… and effluent storage constraints”. This statement does not come with citations, I would assume it was from the authors. However, from the perspective of the membrane process, high permeability would enhance, rather than reduce, membrane fouling due to concentration polarization at the interface between the membrane and the solution. Thus, can the authors explain this statement or provide a proper citation to support this?

Thanks for the very interesting question, regarding the statement, we would like to say that the high permeability of SiC membrane can be beneficial for water treatment in comparison to other membrane material because for a similar fouling rate, the flux applied can be higher which helps to maintain the production value. However, this sentence has been modified and support with a citation in the manuscript. Regarding polarization concentration, due to the filtration operating parameter, a high cross flow velocity and low permeate flux. The diffusion phenomenon is not preponderant in membrane and has low impact the fouling which are mainly pore blocking and cake deposition (no difference in permeate flow after replacing the effluent with water).

  1. For table 2, the operating conditions applied on SiC membranes are not applicable to all water sources (five different effluents), I am confused about the presence of such data here.

Thanks for your comments, to improve the comprehension, we decided to change the format and added sentences in the text which can replace the table 2. However, first objectives of the table was to give an overview of the range of parameters applied during the campaign for all the fluid with no distinction because new conditions were applied on membrane side based on previous results obtained.

  1. For Figure 2(a-e), it is very difficult to compare each figure because both the y and x axis are different. I would suggest the author replot the figure 2 to clearly indicate the difference among effluents. Also, please add distinct colors for all figures in the manuscript.

Thanks for your remark, figures have been modified and presenting in color for better understanding

  1. Line 333 - 334: “Application of a permeate flow…as it is shown in Figure 5”. In RO, a high permeability would result in severe concentration polarization that leads to more membrane fouling. I am curious why you observed differently. Can you explain in more detail why you observed this?

SiC membrane used are not RO membrane and the polarization concentration, which is generate by the diffusion mechanism are not an important on MF or UF membrane. Moreover, the presence of solid particles in feed water and in membrane channels, supposed a fouling mechanism mainly by pores blocking and cake deposit. Thus, increase the flux leads to a higher membrane fouling appeared. From figure 5 b a low permeate flow give more fouling resistance due to the significant decrease in permeability from the beginning of the filtration.

  1. I am curious about what types of fouling that caused the membrane to fail in your experiments. I am aware that you have five effluents with different fouling potential. Can you indicate specifically what kind of fouling (organic, scaling or biofouling) is for each effluent? Throughout the manuscript, I don’t see the authors explain the fouling.

Thank you for your interesting question, it is supposed that all fouling mechanism appears on the membrane surface during the filtration (pore blocking and cake layer at the same time) for each effluent. The main difference between effluent is their concentration, more or less concentrated in salts, particles matter, hydrocarbons, … but their fouling mechanisms are similar. Resistances characterization (organic, inorganic) have been made during the CIP to determine the influence of alkaline and acidic compound in membrane performances recovery. Results showed that most of the water permeability have been recovered thanks to acid batches due to the high proportion of inorganic compounds (MgOH2, metal ions, …). However, without an alkaline batch not the entire water permeability is recovered. Both actions, acidic and alkaline are thus required to perfectly cleaned the membrane. These results validate the fact the organic and inorganic compound composed the fouling layer.

  1. Also, I am curious what are the irreversible and reversible fouling, do you have SEM images? The type of fouling needs to be identified because, in practice, you might need different strategies to handle different fouling. 

Thank you for your interesting question but we have not SEM image first because this was not the objective of our research and then because we wanted to perform all the filtration experiment with only one membrane in order to conclude on the mechanical resistance and cleaning efficiency. In this context, we could not break the membrane to observe the fouling inside the channel. Clearly, the magnesia suspension is the main cause of fouling and has the advantage of almost total dissolution at basic pH. This makes the fouling reversible whatever the location (in the pores or on the surface).